# Neural Embeddings for Nearest Neighbor Search Under Edit Distance

## Abstract

The edit distance between two sequences is an important metric with many applications. The drawback, however, is the high computational cost of many basic problems involving this notion, such as the nearest neighbor search. A natural approach to overcoming this issue is to *embed* the sequences into a vector space such that the geometric distance in the target space approximates the edit distance in the original space. However, the known edit distance embedding algorithms, such as Chakraborty et al. (2016), construct embeddings that are data-independent, i.e., do not exploit any structure of embedded sets of strings. In this paper we propose an alternative approach, which learns the embedding function according to the data distribution. Our experiments show that the new algorithm has much better empirical performance than prior data-independent methods.

## 1 Introduction

The *edit distance* is a popular metric for computing the dissimilarity between two strings. Given strings $x$ and $y$, the edit distance $d_{\mathrm{E}}(x, y)$ between them is equal to the minimum number of character insertions, deletions and substitutions needed to transform $x$ into $y$. The metric and its many variants have been defined and studied since the 1960s (Levenshtein, 1966; Wagner and Fischer, 1974). It has found many applications in computational biology (to measure the similarity between DNA/RNA sequences), natural language processing, information theory and other fields.

From the algorithmic perspective, the main drawback of edit distance is its computational complexity. The best known algorithm for computing the distance between two strings of length $n$ takes time that is roughly quadratic in $n$ (Masek and Paterson, 1980); there is also evidence that this runtime is essentially optimal (Backurs and Indyk, 2015). Furthermore, many applications require finding the closest string in a large collection of (say) $N$ strings. Thus, even if we could estimate the edit distance in time $O(n)$, computing the distances to all strings in the collection would take $O(Nn)$ time, which is infeasible for large data sets.

A natural approach to resolving both issues relies on *low-distortion embeddings*. The idea is to construct a mapping $f$ that maps strings into a $d$-dimensional space $\Re^d$, such that the edit distance $d_{\mathrm{E}}(x, y)$ is approximated by the distance $\|f(x) - f(y)\|$ between the images of the strings, up to some factor $D$ called *distortion*. If the mapping $f$ can be evaluated efficiently, this immediately implies an efficient way to compute the edit distance. Furthermore, embedding strings into space equipped with a norm $\| \cdot \|$ makes it possible to use nearest neighbor search algorithms designed for that norm. For many normed spaces, e.g. Minkowski norms $\ell_p^d$, many efficient approximate nearest neighbor search algorithms are available. This dual goal has motivated a long line of research focused on designing efficient low-distortion embeddings of variants of edit distance into normed spaces (Muthukrishnan and Sahinalp, 2000; Cormode et al., 2001; Ostrovsky and Rabani, 2005; Batu et al., 2006; Jowhari, 2012; Andoni and Onak, 2012; Chakraborty et al., 2016). The method given in the last paper (Chakraborty et al., 2016) was shown to yield good results in practice, in the context of large scale similarity joins, i.e., identifying all pairs of similar sequences (Zhang and Zhang, 2017).

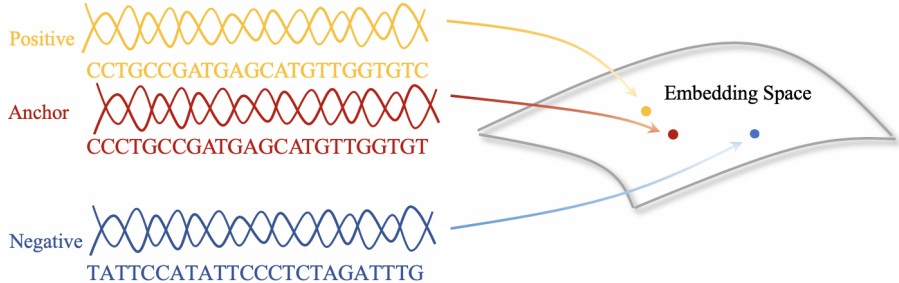

Figure 1: Triplet loss for embedding DNA sequence inputs. Given anchor point $A$, a positive point $P$ that is close to $A$ in terms of edit distance, and a negative point $N$ that is not close to $A$. We hope our embedding function $f$ satisfies $f(A)$ is closer to $f(P)$ than $f(N)$ in $\ell_p$.

A weakness of the aforementioned algorithms is that they are *data-independent*, i.e., they do not exploit any structure of embedded sets of strings. In contrast, over the last decade, a large body of research on "learning to hash" showed that *data-dependent* methods often yield (binary) embeddings with a significantly lower distortion than their data-independent counterparts (Wang et al., 2018). Such methods, however, are typically tailored to obtaining binary representations of vectors that already live in a geometric (typically Hilbert) space. It is thus natural to investigate whether similar benefits of data-dependent methods can be obtained for edit distance embeddings as well.

In this paper we investigate whether the "learning to hash" approach can yield lower distortion embeddings than those obtained using data-independent methods. Since our goal is to obtain distance-preserving mappings of sequences, we use Recurrent Neural Networks as a model implementing those mappings. Specifically, we leverage a two-layer GRU network, depicted in Figure 2 and described in more details in Section 4. The network inputs a sequence of characters (e.g., A, G, T or C for DNA sequences), and outputs a vector of symbols, such that the distance between the vectors approximates the edit distance between the original sequences.

The standard loss function for optimizing the network is the *triplet loss* (see Figure 1), which has two problems in our scenario. The first problem is distance scaling: triplet loss only controls the relative distance ordering among data points after embedding, rather than the absolute embedding distance between any two data points. This prevents us from getting accurate edit distance estimation from the embedding. To alleviate this issue, we designed a warm-start phase (Phase 1) before optimizing triplet loss (Phase 2), which initializes the embedding so that the discrepancy between the original edit distance and the embedding distance is minimized. The second problem is worst case performance of our model. Due to the nature of learning, the network we learned fits the training data better, and may not do well on unseen hard examples. To alleviate this issue, we append a back up phase (Phase 3), which deals with hard cases.

We emphasize that all the three phases are optimizing *the same network*. We run Adam algorithm for each phase until convergence, and then use its final state as the initial state for the next phase. Although in principle one may use a single phase with combination of different loss functions to train our model, this requires additional hyperparameter adjustment for the combining ratios. Our current three phase training does not need additional hyperparameters, and therefore can easily fit different scenarios.

The Phase 3 design is non-trivial. Our starting observation is that Recurrent Neural Networks that we employ are not "compatible" with the data-independent mappings of Chakraborty et al. (2016)(which we call CGK from now on). This is because CGK can generate different number of copies for every input character, which is a discrete decision that is hard to be learned by neural networks. We show, however, that this "extra power" of the algorithm is not necessary. In particular, we introduce a "truncated" version of the CGK algorithm (referred to as CGK'), that outputs only 1 or 2 symbols per round. We then show that CGK' has the same guarantees as the original CGK algorithm. Based on this observation, we design our Phase 3 to be an optimized version of CGK'. Specifically, since each round there will be either 1 or 2 symbols in CGK', we can directly compare the

two cases and use the difference as approximation to the gradients to train our model. Empirically, Phase 3 does not affect the recall on ordinary datasets much, but greatly improves performance on hard instances.

To evaluate the embeddings, we employ them for nearest neighbor search in a collection of sequences, and compare their performance to CGK. As in Zhang and Zhang (2017) we use three data sets: Gen50ks (containing sequences from the human genome project), UniRef (protein sequence dataset from the UniProt project) and Trec (containing textual information from Medline, an online medical information database). After learning the embeddings, we generate a set of queries, and for each query we identify $C$ sequences in the data set (called *candidates*) that are closest to the query *in the embedded space*. We then measure "top-$K$ recall", i.e., the fraction of the *actual* top $K$ nearest neighbors (computed using the exact edit distance) returned as one of the $C$ candidates. Our results demonstrate that, over a wide range of parameters $K$ and $C$, our learned embedding consistently achieves higher recalls than the algorithm of CGK, often by a factor of 3 or more. Similarly, our method often obtains identical recall to that of CGK while using an order of magnitude fewer candidates. This means that in a typical nearest neighbor search "filtering" pipeline, where the candidates are quickly identified in the embedded space and then the exact edit distance is computed between the candidates and queries to identify the final answers, our method reduces the number of exact edit distance computations by up to one order of magnitude. Moreover, our learned embedding also has much more organized structure than CGK, as shown in Appendix A.4.

## 2 Related work

### 2.1 Edit Distance Computing

Exact and approximate algorithms for edit distance computation have been broadly studied, see Navarro (2001) for an overview. For exact algorithms, Wagner and Fischer (1974) gave $O(n^2)$-time algorithm using dynamic programming. This was later improved to $O(n^2/\log n)$ by Masek and Paterson (1980). Faster approximation algorithms were studied as well, see e.g., Landau et al. (1998); Batu et al. (2003); Bar-Yossef et al. (2004); Batu et al. (2006); Andoni et al. (2010); Andoni and Onak (2012). The current state of the art consists of a near-linear algorithm of Andoni et al. (2010) guaranteeing $(\log n)^c$-approximation (with a trade-off between the running time and the exponent $c$), and a constant factor approximation $O(n^{2-2/7})$-time algorithm of Chakraborty et al. (2018). To the best of our knowledge, neither algorithm has been implemented.

### 2.2 Edit Distance Embedding

As mentioned in the introduction, there has been a long line of research focused on designing embeddings of various variants of the edit distance into geometric space. For the edit distance as defined in the introduction, the best known algorithm is due to Ostrovsky and Rabani (2005), who designed an embedding into the Hamming space with distortion $2^{O(\sqrt{\log n \log \log n})}$. This algorithm was quite complex, and to the best of our knowledge has never been implemented. Recently, Chakraborty et al. (2016) gave a substantially different embedding with distortion $O(k)$ where $k$ is the edit distance between the original strings. The empirical evaluation of the algorithm, given in Zhang and Zhang (2017), demonstrated that the embedding yields good results in practice, and that the empirical distortion of this embedding is typically much lower than the worst case bound. Zhang and Zhang (2019) also proposed a novel string partition based algorithm for edit similarity join, without generating embeddings. Gomez-Bigorda et al. (2017) presented a novel approach for word embedding in Levenshtein space to improve downstream handwritten word spotting. Their method is similar to our Phase 1, but our model deals with longer sequences instead of words, and also adopts triplet loss training to capture sequence similarity (Phase 2) and CGK based algorithm to learn sequence alignment (Phase 3).

### 2.3 Similarity Capturing

Neural models are powerful to learn embeddings that capture latent similarity. For example, word2vec (Mikolov et al., 2013) learns semantic and syntactic similarity of a word from its

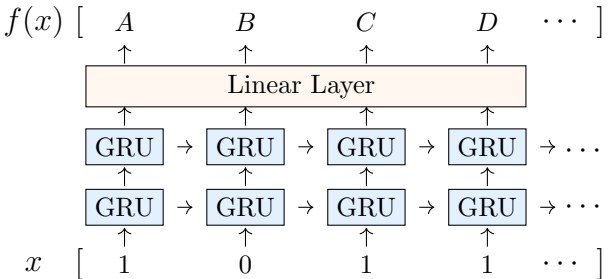

Figure 2: Our embedding model leveraging two-layer GRU.

linguistic context. The idea has been extended to learn sentence or document similarity (Le and Mikolov, 2014), biological sequence similarity (Asgari and Mofrad, 2015; Ng, 2017). The aforementioned methods characterize input representations based on their context. A different approach aims to learn a distance metric, see Kulis (2013) for an overview. Among them, triplet network (Weinberger and Saul, 2009; Hoffer and Ailon, 2015; Schroff et al., 2015) uses triplet loss to learn similarity over triplets. Part of our model gets inspiration from triplet network.

## 3 PRELIMINARIES

We use $\mathcal{D}$ to denote the alphabet, i.e., the set of symbols used in the strings. For any two strings $x, y \in \mathcal{D}^*$ we use $d_{\mathrm{E}}(x, y)$ to denote the edit distance between $x$ and $y$, i.e., the minimum number of symbol insertions, deletions or substitutions needed to transform $x$ into $y$. We define the distance metric $d(x, y) = ||x - y||_2^2$. If the strings have different lengths, we pad the shorter string to make their lengths equal. In the binary case, $d(x, y)$ corresponds to the Hamming distance between $x$ and $y$.

The CGK algorithm is described below. Its input consists of a string $x$ of length $n$ (to be embedded), and a sequence of $6n$ random bits $r$, which are interpreted as a sequence of $3n$ random hash functions $h_1, \cdots, h_{3n} : \{0, 1\} \to \{0, 1\}$. It produces a sequence $f(x, r)$, obtained by replicating each symbol in $x$ at least once (the actual number is a random variable depending on $x$ and $r$).

---

**Algorithm 1** CGK Embedding Function $f$

---

**Input:** $x \in \{0, 1\}^n$, and a random string $r \in \{0, 1\}^{6n}$
**Output:** $f(x, r) \in \{0, 1\}^{3n}$
  Interpret $r$ as the description of $h_1, \cdots, h_{3n} : \{0, 1\} \to \{0, 1\}$.
  Initialization: $i = 1$, $a = (0, \cdots, 0) \in \{0, 1\}^{3n}$
  **for** $j = 1, 2, \cdots, 3n$ **do**
    $a_j = x_i$;
    $i = i + h_j(x_i)$;
  **end for**
  Set $f(x, r) = a$.

---

## 4 LEARNING-BASED EMBEDDING METHOD

Recurrent Neural Network (RNN) models proved effective for sequence embedding tasks like language modeling (Mikolov et al., 2011), image captioning (Mao et al., 2015), protein sequence embedding (Bepler and Berger, 2019), etc. Long Short Term Memory (LSTM) (Hochreiter and Schmidhuber, 1997) and Gated Recurrent Unit (GRU) (Cho et al., 2014) utilize gates to overcome the long-term dependency problem and often yield improved empirical results. Our learning-based embedding model leverages two-layer GRU depicted in Figure 2. The input is a sequence of $n$ characters in one-hot encoding, $x = \{x_1, x_2, ..., x_n\}$. We feed the input sequence $x$ into two-layer GRU and a linear layer to get the embedding sequence $y = f(x)$. Here $\delta_i$ represents the dropout rate in GRU. We use Adam optimizer and 4:1 train:validation split for cross-validation to tune the hyper-parameters.

Phase 1 | Phase 2 | Phase 3

x → Embed → $f_1(x)$

$|d_E - d|$

y → Embed → $f_1(y)$

p → Embed → $f_2(p)$

a → Embed → $f_2(a)$ → triplet loss

n → Embed → $f_2(n)$

$f_3(x)$ $\begin{bmatrix} A & B & C & \cdots \end{bmatrix}$

Add minor?

$f_3'(x)$ $\begin{bmatrix} A\,A & B\,B & C\,C & \cdots \end{bmatrix}$

Figure 3: Our three phase training method: Phase 1 as warm start; Phase 2 learns similarity; Phase 3 for hard case. *Embed* refers to embedding model depicted in Figure 2.

$$h_i^1 = GRU^1(x_i, h_{i-1}^1) \tag{1}$$

$$h_i^2 = GRU^2(h_i^1 \delta_i, h_{i-1}^2) \tag{2}$$

$$y_i = W h_i^2 + b \tag{3}$$

$$y = \{y_1, y_2, ...y_n\} \tag{4}$$

## 4.1 Phase 1: Absolute Loss as Warm Start

We use Phase 1 to tune the embedding such that the absolute difference between the edit distance and the embedding distance is small. In every iteration, we take a pair of input strings $x$ and $y$, and pass it through two embedding networks denoted as $f_1$, which share weights. The loss is defined as:

$$L_1 = |d_E(x, y) - d(f_1(x), f_1(y))| \tag{5}$$

## 4.2 Phase 2: Triplet Loss for Similarity Capturing

Phase 2 initializes the network with weights computed in Phase 1. It leverages triplet loss (Weinberger and Saul, 2009) to capture the dependencies between string similarities. In each iteration we randomly choose an original string as the *anchor* and define positive and negative strings with respect to their edit distances from the anchor. More specifically, We randomly sample the positive pairs from top-5 edit distance and sample negative pairs as the top-30 distance in the embedded space, which is a fairly standard approach for triplet loss (Weinberger and Saul, 2009; Sablayrolles et al., 2019). The anchor, the positive and the negative string form a *triplet*. The triplets are passed through three networks, denoted by $f_2$, which share common weights. For an anchor $a$, a positive string $p$ and a negative string $n$, we formulate the triplet loss as:

$$L_2 = \max(0, d(f_2(a), f_2(p)) - d(f_2(a), f_2(n)) + \text{margin}) \tag{6}$$

Further, we set margin to be $d_E(a, n) - d_E(a, p)$. Intuitively, we hope that the difference between the hamming distance between the positive and the negative pairs is larger than that of the edit distance, namely $d(f_2(a), f_2(n)) - d(f_2(a), f_2(p)) \geq d_E(a, n) - d_E(a, p)$. See Figure 1 for illustration.

## 4.3 Phase 3: Insertion Loss for Hard Cases

Phase 3 is initialized using weights computed in Phase 2. During each iteration a pair of input strings is passed through two embedding networks denoted as $f_3$, which share weights. Let $x$ and $y$ represent the original input strings and $f_3(x)$ and $f_3(y)$ denote their embeddings. Denote the CGK' (a variant of CGK, see Algorithm 3 in the appendix) with random initialization as $g$, and we apply it on $f_3(x)$, $f_3(y)$ and get $f_3'(x) = g(f_3(x))$, $f_3'(y) = g(f_3(x))$. $f_3'$ is our final embedding function, which is essentially a concatenation of an embedding network $f$ and the CGK' algorithm $g$.

Optimization in Phase 3 has two different losses: the absolute loss and the insertion loss. The absolute loss ($L_3$) ensures that the distance of the final embedding is close to the edit distance of $x$ and $y$. The insertion loss is defined as follows. For each character $c$ in $f_3(x)$, we

compare the two cases that with or without duplication of $c$ to optimize the hash functions and the threshold. See Appendix A.1 for more details.

$$L_3 = |d_E(x, y) - d(f'_3(x), f'_3(y))| \tag{7}$$

As we mentioned before, we use CGK' instead of CGK because it restricts the number of replications of each symbol to be at most 2, therefore we can compare the two cases to get an approximation to the gradient, and then optimize the parameters in CGK'. In what follows we show that this modification does not affect the worst-case guarantees of the method (i.e., CGK' with random initialization). See Appendix A.2 for the proof.

**Theorem 1.** *The mapping* $f : \{0,1\}^n \times \{0,1\}^{4n} \to \{0,1\}^{2n}$ *computed by Algorithm 2 satisfies the following properties:*

(1) *Given* $r \in \{0,1\}^{4n}$ *and* $f(x,r) \in \{0,1\}^{2n}$ *as the input, we can always decode* $x$.

(2) *For every* $x, y \in \{0,1\}^n$, $d_E(x,y)/2 \le d(f(x,r), f(y,r))$.

(3) *There exists a constant* $c_0$ *such that for every positive constant* $c \ge c_0$ *and every* $x, y \in \{0,1\}^n$, $d(f(x,r), f(y,r)) \le c \cdot d_E(x,y)^2$ *with probability at least* $1/2$.

## 5 EXPERIMENTS

### 5.1 DATASETS

We evaluate our neural embedding with three datasets as in Zhang and Zhang (2017). Specifically, we use **Gen50ks**: a genetic sequence dataset from the human genome project; **UniRef**: UniRef90 protein sequence dataset from the UniProt project; and **Trec**: a reference dataset from Medline (an online medical information database). See Table 1 for their properties. Furthermore, following Zhang and Zhang (2017) we truncate some input strings to avoid dealing with very long outlier strings. For Gen50ks dataset, the string lengths sharply concentrate around 5000, so we set the truncation length to be 5000 and thresholding affects the distances only minimally. For UniRef and Trec dataset, string lengths vary drastically. In order to avoid biasing the original data distribution, we set a rather large threshold(1500 and 2500) so that only a small proportion(1.16% and 0.97%) of the strings exceed the threshold. For Gen50ks, UniRef and Trec dataset, we train our model with 1000, 2000, 2000 strings (including base and query) respectively, which means that we train with less than 2% data points in the whole dataset.

Table 1: Statistical analysis of three datasets.

| Dataset | Gen50ks | UniRef | Trec |
|---|---|---|---|
| Alphabet Size | 4 | 25 | 37 |
| Dataset Size | 50000 | 400000 | 233435 |
| Avg Len. | 5000 | 445 | 1217 |
| Max Len. | 5152 | 35213 | 3947 |
| Min Len. | 4829 | 200 | 80 |

### 5.2 NEAREST NEIGHBOR SEARCH

#### 5.2.1 EXACT NEAREST NEIGHBOR SEARCH

We evaluate our model in the context of nearest neighbour search. We randomly select 100 queries and use the remainder of the dataset as the base set. Our goal is to find strings in the base set most similar to the query strings. To this end we use linear scan to identify the $C$ sequences in the data set (called *candidates*) that are closest to the query *in the embedded space*. We compare embeddings generated by our neural model, CGK and its close variant CGK', as well as the identity embedding (which is optimal if no insertions or deletions are needed). Since CGK and its variants are randomized, we measure the average of five independent runs.

For all three datasets we observe that our neural model offers significant performance gains compared to the other embedding methods.

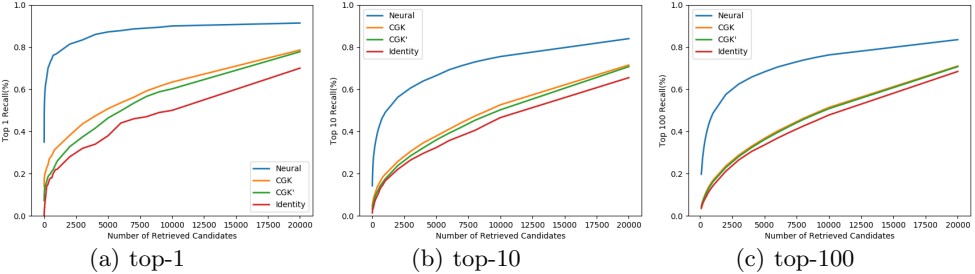

Figure 4: Gen50ks: top $K$ recalls for our algorithm (Neural) vs. CGK, CGK' and identity, as a function of the number of candidates.

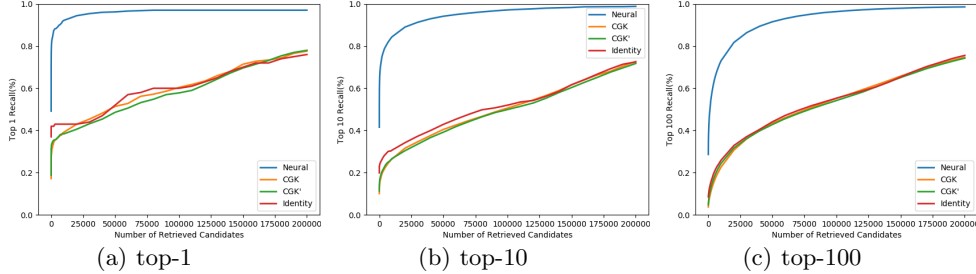

Figure 5: Uniref: top $K$ recalls for our algorithm (Neural) vs. CGK, CGK' and identity, as a function of the number of candidates.

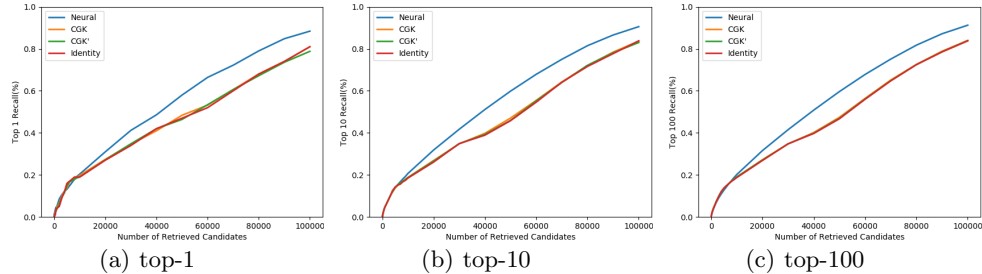

Figure 6: Trec: top $K$ recalls for our algorithm (Neural) vs. CGK, CGK' and identity, as a function of the number of candidates.

### 5.2.2 LSH-based Approximate Nearest Neighbor Search

In the previous section we used linear scan to find $C$ nearest neighbors in the embedded space. Although this task can be often accomplished efficiently (e.g., using GPUs), in many scenarios more efficient approximate nearest neighbor search algorithms need to be employed. In this section we investigate one such instantiation employing Locality-Sensitive Hashing (LSH). We note that the same framework was used in Zhang and Zhang (2017) to evaluate CGK.

We use the cross-polytope LSH family under the Euclidean distance with multi-probe LSH (Andoni et al., 2015). For genetic dataset Gen50ks, we build 50 hash tables and 10-bit hash functions(We vary the number of hash bits and find 10-bit hash functions give the best performance). We test with five randomly selected 100 queries and record the mean and standard deviation as in Table 2.

Table 2: Top-1 Recall for Gen50ks using LSH

| Number of Probes | 50 | 100 | 200 | 400 | 800 | 1600 | 3200 |
|---|---|---|---|---|---|---|---|
| **Neural** | **50.2±2.7** | **63.6±1.8** | **74.0±1.6** | **84.2±2.8** | **93.0±3.5** | **98.6±1.3** | **99.8±0.4** |
| CGK | 18.0±2.9 | 23.6±2.8 | 34.4±2.0 | 47.6±3.7 | 68.6±5.2 | 88.8±3.4 | 98.2±1.3 |
| CGK' | 11.0±2.6 | 16.8±2.2 | 26.0±3.0 | 43.0±3.8 | 65.8±4.6 | 83.6±3.4 | 98.0±0.6 |
| Identity | 4.6±0.4 | 9.8±1.6 | 19.2±1.7 | 33.2±4.0 | 56.8±3.7 | 83.4±2.6 | 97.4±0.8 |

Table 3: Recall for Gen50ks, edit distance join threshold K = 50

| Number of Probes | 50 | 100 | 200 | 400 | 800 | 1600 | 3200 |
|---|---|---|---|---|---|---|---|
| **Neural** | **91.6** | **96.4** | **98.3** | **99.5** | **100.0** | **100.0** | **100.0** |
| CGK | 79.6 | 83.9 | 86.6 | 90.6 | 94.5 | 97.6 | 99.7 |
| CGK' | 65.0 | 69.5 | 74.4 | 80.3 | 88.0 | 94.8 | 99.0 |
| Identity | 7.9 | 12.9 | 22.0 | 38.6 | 60.2 | 81.5 | 95.7 |

Table 4: Recall for Gen50ks, edit distance join threshold K = 100

| Number of Probes | 50 | 100 | 200 | 400 | 800 | 1600 | 3200 |
|---|---|---|---|---|---|---|---|
| **Neural** | **82.5** | **91.0** | **96.4** | **99.0** | **99.8** | **99.9** | **100.0** |
| CGK | 61.9 | 66.9 | 72.9 | 79.8 | 88.1 | 95.0 | 99.2 |
| CGK' | 42.6 | 48.5 | 55.5 | 65.3 | 78.6 | 91.8 | 98.8 |
| Identity | 6.3 | 11.1 | 20.6 | 36.7 | 59.5 | 82.8 | 96.6 |

Table 5: Ablation Study: Top-1 Recall for Gen50ks

| Candidate Num | 1 | 10 | 50 | 80 | 100 | 200 | 500 | 800 | 1000 | 2000 | 5000 |
|---|---|---|---|---|---|---|---|---|---|---|---|
| Neural Phase 2+3 | 26.4 $\pm$3.4 | 40.6 $\pm$4.9 | 48.2 $\pm$4.1 | 51.8 $\pm$3.4 | 53.0 $\pm$3.5 | 59.8 $\pm$3.4 | 67.8 $\pm$3.7 | 71.0 $\pm$3.1 | 74.2 $\pm$3.3 | 79.4 $\pm$3.7 | 86.2 $\pm$1.3 |
| Neural Phase 1+3 | 7.6 $\pm$0.4 | 14.8 $\pm$1.1 | 19.4 $\pm$1.0 | 22.0 $\pm$2.9 | 24.4 $\pm$3.0 | 28.2 $\pm$1.1 | 34.4 $\pm$1.9 | 40.0 $\pm$3.8 | 42.8 $\pm$3.4 | 51.8 $\pm$4.0 | 63.6 $\pm$4.4 |
| Neural Phase 1+2 | **36.6** $\pm$**1.3** | 49.8 $\pm$3.1 | 60.6 $\pm$1.9 | **63.2** $\pm$**1.7** | **65.0** $\pm$**1.8** | **70.2** $\pm$**2.7** | **77.4** $\pm$**2.8** | **81.2** $\pm$**1.8** | **82.4** $\pm$**1.8** | **88.8** $\pm$**2.4** | **94.2** $\pm$**1.4** |
| Neural Phase 1+2+3 | 36.2 $\pm$1.8 | **53.4** $\pm$**3.8** | **60.8** $\pm$**3.7** | 62.8 $\pm$2.9 | 64.0 $\pm$3.0 | 66.8 $\pm$2.9 | 71.8 $\pm$1.7 | 75.2 $\pm$2.1 | 76.4 $\pm$2.6 | 81.8 $\pm$1.3 | 87.0 $\pm$1.5 |

## 5.3 String Similarity Join

Zhang and Zhang (2017) adopts the vanilla bit-sampling LSH for hamming distance, which does not suit our embedding as the neural embedded strings are continuous values instead of discrete characters. So for our string similarity join experiments, we continue to use the cross-polytope LSH family under the Euclidean distance with multi-probe LSH.

The LSH table parameters are the same with the approximate nearest neighbor setting. We run PassJoin (Li et al., 2011) to evaluate our join results.

## 5.4 Ablation Study

We further do ablation study to evaluate the function of three phases in our model. We notice a performance drop by removing Phase 1 or Phase 2. We explain this as Phase 1 provides good initialization which makes similarity learning easier while Phase 2 learns string similarity. Removing phase 3 generally does not lower the performance of top-1 recall much. This might be that these datasets do not contain many hard instances(like coexisting small shift and small substitution). Phase 3 does show its effectiveness in our synthetic hard case dataset, see Appendix A.3. We further show the standard deviation by testing with five different query sets, each of which contains 100 randomly selected strings.

## 6 Conclusion

In this paper, we introduce the idea of using neural embedding for computing nearest neighbor search under edit distance. Our method has three phases, i.e., the warm-start phase for generating an initial network with the correct scale, the triplet loss phase for learning relative distance among data points, and the back up phase for dealing with hard cases. Our last phase is non-trivial, because we designed a truncated version of CGK algorithm (aka CGK') and proved that it has worst case performance guarantee. Our phase 3 directly optimizes CGK' and easily solves hard cases. Overall, our method gives much better empirical results than the existing methods for nearest neighbor search.

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

# A  Appendix

## A.1  CGK' algorithm and Phase 3

In Algorithm 2, we present the truncated version of CGK that we use in the paper. Compared with Algorithm 1, the main difference is in CGK', every character will be duplicated for at most twice (with probability 0.5). Our Phase 3 is an optimized version of CGK'. We use Algorithm 3 in Phase 3. The main difference between Algorithm 2 and 3 is that we are using a threshold variable *thres* in the if-statement, to decide whether to duplicate the character or not. Moreover, the if-condition is evaluated based on the inner product between hash function $h_j$ and a one hot vector $x_i$. Hence, Algorithm 3 is just a more general version of Algorithm 2.

Then how do we optimize the parameters in Algorithm 3(E.g., *thres* and *h*)? Since Algorithm 3 involves discrete decisions with if-statements, we manually define the gradients in Algorithm 4, by comparing the case whether the character is duplicated or not and calculate how it affects the loss.

---

**Algorithm 2** CGK' Embedding Function $f$

---

**Input:** $x \in \{0,1\}^n$, and a random string $r \in \{0,1\}^{4n}$
**Output:** $f(x,r) \in \{0,1\}^{2n}$
  Interpret $r$ as the description of $h_1, \cdots, h_{2n} : \{0,1\} \to \{0,1\}$.
  Initialization: $j = 1, a = (0, \cdots, 0) \in \{0,1\}^{2n}$
  **for** $i = 1, 2, \cdots, n$ **do**
    **if** $h_j(x_i)$ **then**
      $a_j = x_i$;
      $j = j + 1$;
    **else**
      $a_{j+1} = a_j = x_i$;
      $j = j + 2$;
    **end if**
  **end for**
  Set $f(x,r) = a$.

---

**Algorithm 3** Learning-Augmented Embedding Function $f'$

---

**Input:** $x$ where $x_i$ is a one-hot vector, $H$ is a $2 \times 2n$ matrix interpreted as $h_1, ..., h_{2n}$, *thres* $\in R$ after optimization
**Output:** $f'(x)$, set *Duplicate* which records the duplicated bits in $f'(x)$
  Initialization: $j = 1, a = \mathbf{0}_{2 \times 2n}, Duplicate = \emptyset$
  **for** $i = 1, 2, \cdots, n$ **do**
    **if** $h_j \cdot x_i < thres$ **then**
      $a_j = x_i$;
      $j = j + 1$;
    **else**
      $a_{j+1} = a_j = x_i$;
      $Duplicate \leftarrow Duplicate \bigcup \{j\}$
      $j = j + 2$;
    **end if**
  **end for**
  Set $f'(x) = a$.

---

## A.2  Proof of the Main Theorem

For simplicity, denote the two embedded strings $f(x,r), f(y,r)$ as $f^x, f^y$. Below we always use $i$ to denote the index for the input strings, and use $j$ to denote the index for the embedded strings. We use $\text{idx}(x, j)$ to denote the corresponding index $i$ for producing the

---

**Algorithm 4** Optimization Scheme for $h$ and *thres* in Algorithm 3

---

**Input:** $x, y, f'(x), f'(y)$, $H$ interpreted as $h_1, ..., h_{2n}$, *thres*, set *Duplicate* in Algorithm 3,
learning parameters $\epsilon, \eta$
**Output:** $H$ and *thres* after optimization
  Initialization $j = 1$
  $L = |d_H(f'(x), f'(y)) - d_E(x, y)|$;
  **for** $i = 1, 2, \cdots, n$ **do**
    $g(x) = f'(x)$
    **if** $j \in Duplicate$ **then**
      Remove $g(x)_{j+1}$ from $g(x)$;
      $L' = |d_H(g(x), f'(y)) - d_E(x, y)|$;
      **if** $L' < L$ **then**
        $thres = thres + \epsilon$;
        $h_j = h_j - \eta \cdot x_i$;
      **end if**
      $j = j + 2$;
    **else**
      Duplicate $g(x)_j$ in $g(x)$;
      $L' = |d_H(g(x), f'(y)) - d_E(x, y)|$;
      **if** $L' < L$ **then**
        $thres = thres - \epsilon$;
        $h_j = h_j + \eta \cdot x_i$;
      **end if**
      $j = j + 1$;
    **end if**
  **end for**

---

$j$-th index of output $f(x, r)$. Define $\Delta_j \triangleq \mathrm{idx}(x, j) - \mathrm{idx}(y, j)$ as the input index difference between $x$ and $y$ at embedding index $j$.

In Algorithm 2, given the input $x$ and index $i$, we may append one or two copies of $x_i$ into the output. We call the first copy the **major** copy, and the second copy the **minor** copy. The major copy always exists in the output, but the minor copy exists with probability $1/2$, depending on the hash function $h_j$. For any index $j$, if $f_j^x$ and $f_j^y$ are both major copies or both minor copies, we say $j$ is **aligned**. Otherwise, $j$ is **not aligned**. Initially, $f_1^x$ and $f_1^y$ are major copies, therefore index 1 is aligned. See figure 7 for illustration.

We scan $f^x$ and $f^y$ from left to right, and for every index $j$ such that $j+1$ is not aligned but $j$ is aligned, we find the next aligned index $j'$, and group all the indices $j+1, j+2, \cdots, j'-1$ as a sub-index string for the aligned index $j$. For the corner case that $j'$ does not exist, we pick $j'$ as the minimum final $j$ value for computing $f^x$ and $f^y$ (So starting from $j'$, $f^x$ or $f^y$ will be padded with zeros).

Now we focus on the aligned indices. For an aligned index $j$ without a sub-index string, we know index $j+1$ is also aligned, otherwise $j+1$ will be inside the sub-index string for index $j$. In this case, we can show $\Delta_{j+1} - \Delta_j = 0$ by discussing whether index $j+1$ has major or minor copies.

On the other hand, consider an aligned index $j$ with a sub-index string. In this case, we know both $f_j^x$ and $f_j^y$ are major copies. Without loss of generality, we may assume $f_{j+1}^x$ is a major copy and $f_{j+1}^y$ is a minor copy (the other case is symmetric). We have the following lemma for a few facts related to the sub-index string.

**Lemma 2.** *Denote the next aligned index by $j' > j+1$. Assume the sub-index string has length $t \triangleq j' - j - 1 > 0$. Given $T$ such that $T/2 \in \mathbb{Z}^+$, we have*

*(1) $f_{j'}^x$ and $f_{j'}^y$ are major copies,*

*(2) $t \leq T$ holds with probability $1 - 2^{-T}$,*

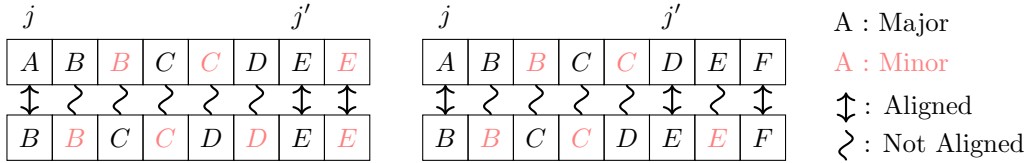

Figure 7: Two possible cases of the sub-index string

*(3) Conditioned on $t \leq T$,*

$$\Delta_{j'} - \Delta_j = \begin{cases} 1, & \text{w.p. } 2/3 \\ 0, & \text{w.p. } 1/3 \end{cases},$$

*Proof.* Since $j'$ is the first aligned index after $j$, we know that for any $j < j_o < j'$, $j_o$ is not aligned, therefore one of $f_{j_o}^x$ and $f_{j_o}^y$ is major and the other one is minor. By Algorithm 2, for the string with the minor copy, with probability 1, the next copy on $j_o + 1$ is a major copy; and for the string with the major copy, the next copy on $j_o + 1$ is a major copy or minor copy with equal probability. However, if both copies on $j_o + 1$ is major, we know $j_o + 1$ is aligned, and thus $j' = j_o + 1$. In other words, this case only happens for the index $j_o = j' - 1$, and for other indices, the two strings are not aligned, and have major copies in turn. That already proves (1).

Since the two strings have major copies in turn, and only major copies will increment the value of $\mathrm{idx}(x, j)$, we know

$$\Delta_{j'} - \Delta_j = \begin{cases} 1, & \text{if } t \text{ is odd} \\ 0, & \text{otherwise} \end{cases},$$

See Figure 7 for the two possible cases. Moreover, every step from $j + 1$ to $j'$ holds with probability $1/2$, so the probability that the sub-index string has length $t$ is $2^{-t}$. By summing over the probabilities of $t$ from 1 to $T$, we proved (2). Finally, (3) holds by comparing the probabilities of $t \leq T$ being odd and being even, which is exactly $2 : 1$. □

**Remark 1.** *Based on Lemma 2(2), if there are $l$ sub-index strings, and for a given $T$ (to be determined later), the probability that all sub-index strings are within length $T$ is bounded by $1 - l \cdot 2^{-T}$ using union bound. Below we discuss the case conditioned on this event.*

Lemma 2(3) states that if an aligned index $j$ has a sub-index string, and $f_{j+1}^x$ is a major copy, then $\Delta_j$ will be incremented by 1 with probability $2/3$, and stay unchanged with probability $1/3$. Since $f_j^x, f_j^y$ are major copies, we know that this case happens with probability $1/4$. The symmetric case where $f_{j+1}^x$ is a minor copy and $f_{j+1}^y$ is a major copy also happens with probability $1/4$.

Combine the two possible cases for aligned index $j$ (with or without sub-index string), and assuming the next aligned major index is $j'$. If $f_j^x$ and $f_j^y$ are minor copies, or are the same, we have $\Delta_{j'} - \Delta_j = 0$ by simple case analysis. Otherwise, we have:

$$\Delta_{j'} - \Delta_j = \begin{cases} 1, & \text{w.p. } 1/6 \\ 0, & \text{w.p. } 2/3 \\ -1, & \text{w.p. } 1/6 \end{cases}, \tag{8}$$

This is because for the two aligned major copies at index $j$, with probability $1/2$, the next index $j + 1$ is still aligned, and as a result $\Delta_j$ does not change. With probability $1/2$, there exists a sub-index string for index $j$, but conditioned on this, the probability that $\Delta_j$ does not change is $1/3$. So the total probability for being unchanged is $1/2 + 1/6 = 2/3$. The other two cases are symmetric, so each has probability $(1 - 2/3)/2 = 1/6$. This defines a random walk process for $\Delta_j$, and the remaining analysis is similar to the analysis in Chakraborty et al. (2016).

Assume that $d_{\mathrm{E}}(x,y) = k$. Let $x = x^{(0)}, x^{(1)}, \cdots, x^{(k)} = y$ be a series of strings such that $d_{\mathrm{E}}(x^{(l-1)}, x^{(l)}) = 1$. Let $i_l$ be the first index on which $x^{(l-1)}$ and $x^{(l)}$ differ. For simplicity we assume that $i_1 < \cdots < i_k$ (and we will later show this is without loss of generality).

Let $d_j$ be the difference between deleted and inserted bits between $x$ and $y$ before the index $j$. Therefore, $d_0 = 0$, and

$$
d_j = \begin{cases} d_{j-1}, & \mathrm{idx}(x,j) \notin \{i_1, \cdots, i_k\} \\ d_{j-1} - 1, & \mathrm{idx}(x,j) = i_l, \mathrm{idx}(x,j-1) \neq i_l, x^{(l)} \text{ is obtained by deletion} \\ d_{j-1} + 1, & \mathrm{idx}(x,j) = i_l, \mathrm{idx}(x,j-1) \neq i_l, x^{(l)} \text{ is obtained by insertion} \\ d_{j-1} & \text{otherwise} \end{cases} , \quad (9)
$$

$d_j$ denotes the required length of "shift" between $f^x$ and $f^y$. In other words, for $j$ such that $\mathrm{idx}(x,j) \in (i_l, i_{l+1})$, and $\mathrm{idx}(x,j) - \mathrm{idx}(y,j) = \Delta_j = d_j$, we know $f_j^x = f_j^y$. Moreover, for any $j' \geq j$ such that $\mathrm{idx}(x,j') \in [\mathrm{idx}(x,j), i_{l+1})$, we know that $f_{j'}^x = f_{j'}^y$ holds as well, because the same hash function will be applied to both strings after $\mathrm{idx}(x,j)$.

By modeling $\Delta_j$ and $d_j$ as a chasing game between a cat and a dog with a kennel, the following lemma was proved in Chakraborty et al. (2016).

**Lemma 3** (Lemma 4.4 in Chakraborty et al. (2016)). *If $d_{\mathrm{E}}(x,y) = k$, then*

$$
\Pr(|\{j : f_j^x \neq f_j^y, j \text{ is an aligned major index}\}| \leq l) \geq \sum_{t=0}^{l} q(t,k)
$$

*Where $q(t,l)$ denotes the probability that the random walk process defined by (8) starting at the origin, reaches the point $l$ at time $t$ for the first time.*

As discussed in Chakraborty et al. (2016), the assumption that $i_1, \cdots, i_k$ are strictly monotone is without loss of generality, because it suffices to modify the updating rule of $d_j$ in (9) such that $d_j$ and $d_{j-1}$ can be differed by more than 1 unit. Once $d_j$ is properly defined, Lemma 3 still applies.

The sum $\sum_{t=0}^{l} q(t,k)$ in Lemma 3 is easy to bound:

**Lemma 4.** *For any $k, l \in \mathbb{N}$, with probability more than $1 - e^{-\frac{l}{24}}$, it holds that*

$$
\sum_{t=0}^{l} q(t,k) \geq 1 - 12k\sqrt{\frac{6}{l}}
$$

*In particular, $\sum_{t=0}^{7776k^2} q(t,k) \geq \frac{2}{3}$.*

*Proof.* Below is a well known fact about random walks that can be found e.g. (Theorem 2.17 in Levin and Peres (2017)). For any $k, l \in N$, it holds that:

$$
\sum_{t=0}^{l} r(t,k) \geq 1 - \frac{12k}{\sqrt{l}}
$$

where $r(t,k)$ denotes the probability a simple random walk starting at the origin reaches point $k$ for the first time in $t$ steps. Let m be the number of non-lazy steps during a t-step random walk defined in (8). By applying *Chernoff Bounds*, for any $0 < \delta < 1$ we have:

$$
P(m > (1-\delta)\frac{1}{3}t) > 1 - e^{-\frac{\delta^2}{6}t}
$$

Therefore, with probability more than $1 - e^{-\frac{\delta^2}{6}t}$, we are performing more than $(1-\delta)\frac{1}{3}t$-steps simple random walk. Setting $\delta = \frac{1}{2}$ gives us the result in Lemma 4. $\square$

Now we are ready to prove our Theorem 1.

*Proof of Theorem 1.* (1) is trivially true, because we can simulate Algorithm 2 using the random string $r$ to decide for each index of $f^x$ whether it is a major copy or not.

To see (2) is true, we need to do case analysis. We scan $f^x$ and $f^y$ from left to right, and skip all indices that are the same. For the first different index $j$, there are three cases:

1. $j$ is aligned major, and $j + 1$ is aligned minor.

2. $j$ is aligned major, and $j + 1$ is also aligned major.

3. $j$ is aligned major, and $j + 1$ is not aligned.

For the first two cases, it suffices to replace the bit at $\text{idx}(x, j)$ of $x$, and we can keep scanning starting from the next aligned major. For the third case, we can do induction on the length $t$ of the sub-index string for $j$, to show that it suffices to make at most $s$ edits if there are $s$ different bits from $j$ to $j + t$. Without loss of generality, we assume that the index $j + 1$ of $f^x$ is a major copy while $j + 1$ of $f^y$ is a minor copy.

- If $t = 1$, which means $j + 2$ is aligned major. In this case, if bits at index $j + 1$ of $f^x$ and $f^y$ are matched, it suffices to edit the bit at index $j$. Otherwise, it suffices to delete $\text{idx}(x, j)$ of $x$, and replace the bit at index $\text{idx}(x, j + 1)$ of $x$ to match $y$.

- If $t = 2$, which means $j + 3$ is aligned major, and the index $j$ of $f^y$ is a major copy. In this case, we check whether the bits at $j + 2$ for $f^x, f^y$ are the same.

  - If yes, it means, $\text{idx}(y, j + 2)$ of $y$ and $\text{idx}(x, j + 2)$ of $x$ are the same. In this case, it suffices to replace $\text{idx}(x, j)$ of $x$ to match $y$.
  - If no, it suffices to replace both $\text{idx}(x, j), \text{idx}(x, j + 1)$ of $x$ to match $y$.

- If $t > 2$, we also check whether the bits at $j + t$ for $f^x, f^y$ are the same.

  - If yes, it means, $\text{idx}(y, j + t)$ of $y$ and $\text{idx}(x, j + t)$ of $x$ are the same. Then we may reduce the analysis to the $t - 2$ case. By induction, we know that from $j$ to $j + t - 2$, we can make the proper edit to match $x$ and $y$.
  - If no, it means $\text{idx}(y, j + t)$ of $y$ and $\text{idx}(x, j + t)$ of $x$ are different. We make one edit on $\text{idx}(x, j + t)$ to match $\text{idx}(y, j + t)$ of $y$, and then reduce the analysis to the $t - 2$ case.

  Notice that in the above two sub-cases, we did not analyze the case at index $j + t - 1$. This is because this index is not aligned, and the minor copy is included in the analysis of index $j$ to $j + t - 2$, while the major copy is included in the analysis of index $j + t$.

The above analysis shows that it suffices to make at most $d(f^x, f^y)$ edits to match $x$ and $y$ until one of $f^x$ or $f^y$ starts to pad zeros. Since there are at most $d(f^x, f^y)$ deletions or insertions on $x$, and originally $x$ and $y$ have the same length, we know that there are at most $d(f^x, f^y)$ padded zeros. Combine the two cases together, we have $d_{\mathrm{E}}(f^x, f^y)/2 \leq d(f^x, f^y)$.

(3) Apply Lemma 3, we know that with probability $(1 - e^{-\frac{l}{24}}) \cdot (1 - 12k\sqrt{\frac{6}{l}})$, there are at most $l$ aligned major indices that satisfies $f_j^x \neq f_j^y$. Notice that, if for any indices, if $f_j^x = f_j^y$, it does not affect $d(f^x, f^y)$. Hence below we consider other cases when $f_j^x \neq f_j^y$:

- If $j$ is an aligned minor, we know that $j - 1$ is aligned major such that $f_j^x \neq f_j^y$, and also $j + 1$ is an aligned major index. Thus the number of such indices is at most $l$.

- If $j$ is not aligned, we know that $j$ belongs to a sub-index string for an aligned major index $j' < j$. Notice that $f_j^x \neq f_j^y$, otherwise $j'$ won't have sub-index string. We bound the number of such indices below.

To recap, we need to bound the expected sum of lengths of sub-index strings that start from an aligned major index $j$ such that $f_j^x \neq f_j^y$. Since the number of such strings is $l$, and the expected length of each string is at most 2, it follows that the expected sum of lengths is at most $2l$. We can thus use Markov inequality to show that this quantity is at most $20l$ with probability at least $9/10$.

We note that in Lemma 3, we did not consider the comparison between $f^x$ and $f^y$ when one of them has padded zeros. However, it is easy to see that in this case every edit operation can only incur one unit of cost in $d(f^x, f^y)$, which is not the worst case.

Therefore, with probability at least $(1 - e^{-\frac{l}{24}}) \cdot (1 - 12k\sqrt{\frac{6}{l}}) - 0.1$, the hamming distance is at most $20l$. Let $l = \frac{ck^2}{20}$, we know the probability is at least

$$(1 - e^{-\frac{l}{24}}) \cdot (1 - 12k\sqrt{\frac{6}{l}}) - 0.1$$

$$= (1 - e^{-\frac{ck^2}{480}}) \cdot (1 - \frac{12\sqrt{120}}{\sqrt{c}}) - 0.1$$

$$\geq (1 - e^{-\frac{c}{480}}) \cdot (1 - \frac{12\sqrt{120}}{\sqrt{c}}) - 0.1$$

The last inequality holds because $k \geq 1$, otherwise the hamming distance is 0. Therefore it suffices pick $c_0 = 50000$, and the expression above is larger than 0.5. Notice that we did not carefully tune the constants here.

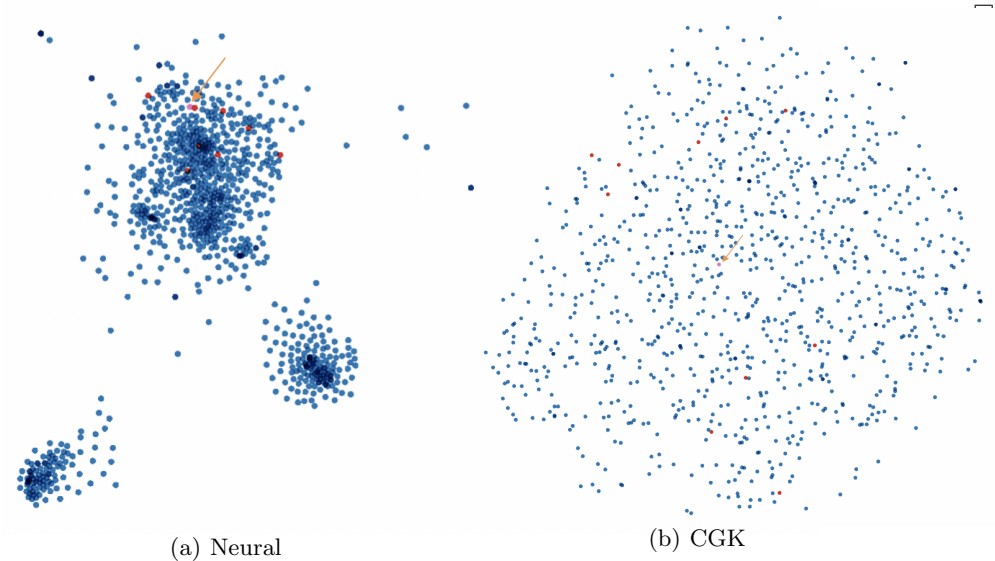

(a) Neural

(b) CGK

Figure 8: t-SNE visualization of neural embedding and CGK embedding. The purple point with orange arrow represents the query string, and the red points represent top-10 edit distance nearest neighbors of the query strings, and the blue points represent the others.

## A.3 Synthetic Data Set

Although Phase 1+2 has good performance for most data sets, we can easily construct synthetic data sets for which it will fail. For example, given a string $x$, applying 1-bit shift to the right gives rise to a large hamming distance but the edit distance is only two. If this 1-bit shift coexists with a 3-bit substitution (so that the hamming distance and edit distance are both equal to three), it would be quite "hard" for Phase 1+2 to find nearest neighbors under edit distance. However, adding Phase 3 easily solves the problem, and gives improved accuracy on the challenging hard cases.

Table 6: Top-1 Recall for the Synthetic Dataset. Candidate number is one.

| Model | Top-1 Recall(%) |
|---|---|
| Phase 1+2+3 | **64.3** |
| Phase 1+2 | 40.9 |
| CGK | 60.1 |
| CGK' | 58.0 |
| Identity | 40.6 |

We have generated a synthetic data set to illustrate the algorithm effectiveness. First we have chosen some prefix strings from the Gen50ks dataset as the query set. Then, for each query, we generate its small shifts (one or two) and small substitution strings (three or four), which are used as the base set. For each query we want to find the closest edit distance string in the base set. The performance of all studied algorithms on this data set are shown in Table 6. It can be seen that adding Phase 3 improves the performance of our method.

A.4    VISUALIZATION

We visualize the embedding results for both CGK and our model using t-SNE (van der Maaten and Hinton, 2008). For CGK embedding, the points distribute over the 2D space randomly no matter what hyper-parameters are used. For neural embedding, on the other hand, points are well clustered. Figure 8 shows 2D t-SNE projections with perplexity 8 and learning rate 10.

