# OpenReview forum: "Neural Embeddings for Nearest Neighbor Search Under Edit Distance"
_ICLR.cc/2020/Conference — Reject_

### Official Review · AnonReviewer1 · 2019-10-22
**Official Blind Review #1**

**Rating:** 6

**Review:**


The paper proposes a scheme to learn representations for sequences
using neural networks such that the Hamming distance in the embedded
space is close to the edit distance in the original representation of
the sequences. This is achieved through a 3-phase algorithm which are
clearly motivated and technically sound. The empirical results clearly
demonstrate the gains from the representation learning on multiple
data sets and scenarios.

Given the clarity, technical soundness and the improved empirical
performance, I am leaning towards an accept. However, there are a
couple of open questions that, if addressed, would help me better
understand the contributions:

- The main concern is the need for the separated out phases instead
  of directly minimizing some combinations of these different
  terms. Is there any inherent reason why the loss function cannot be
  combined to just train a single neural network in an end-to-end
  manner instead of this phase-wise manner. The presence of these
  distinct phases require the reader to figure out when to switch
  phases, and how dependent the downstream performance is to the
  phase change decision.
- As expected, there is a cost to Phase 3. The text is Sec. 5.4 does
  not appear to match up with the Table 5. The table indicates that
  removing phase 3 actually improves performance in all but the top-10
  and top-200 case. The text says something different. Please clarify.
- Moreover, if there is a cost to Phase 3, can this be properly
  explained? If there is a cost to it, could there be a way to decide
  if/when we require the phase 3?


Minor:

- It would also be important to understand the effect of the network
  architecture on the downstream performance. Why is a 2 layer GRU
  necessary and/or sufficient? Some intuition regarding this would be
  very useful and can indicate the ease of applicability of the
  proposed scheme across different data sets.


**Experience Assessment:**

I have read many papers in this area.

**Review Assessment: Checking Correctness Of Derivations And Theory:**

I assessed the sensibility of the derivations and theory.

**Review Assessment: Checking Correctness Of Experiments:**

I carefully checked the experiments.

**Review Assessment: Thoroughness In Paper Reading:**

I read the paper at least twice and used my best judgement in assessing the paper.

---

> ### Author Response · Authors · 2019-11-10
> **Response to Official Review #1**
>
> Thanks for your supportive feedback! We have updated our paper accordingly.
>
> Q: Why use 3 phases, not just a single phase?
> A: We have tried to train phase 1 and phase 2 together as a multi-task problem, but the empirical results were not as good. This is because phase 1 first scales and approximates the edit distance then phase 2 further captures the relative similarity, so the order of phase 1 and phase 2 matters. Besides, phase 3 cannot be jointly trained with phase 1 and phase 2, because in phase 3 we are also optimizing the CGK’ algorithm, which uses approximation to the gradients (see Algorithm 4). This phase is more like a fine tuning step.
> Moreover, even if all phases can be trained together, one needs to tune the final loss function that combines all the losses together. Our current three phases training avoids this tuning problem because we just train the network until convergence for each phase, without using any hyperparameters. So it should be easier (rather than harder) for the readers to use, compared with single phase training.
>
> Q: Table 5 and text in Sec 5.4 are not matched?
> A: Thank you for pointing it out. Here we only show the top-1 recall table (Table 5). We also evaluate on top-10, top-50 and top-100 and have not listed these tables due to limited space.
>
> Q: Can the cost to Phase 3 properly explained? Could there be a way to decide if/when we require phase 3?
> A: Phase 3 essentially involves padding additional characters into the embedding, so that misaligned strings will become aligned. Consider  ‘ACTCGC’ and ‘CACTCG’. ‘CACTCG’ can be edited by shifting one character to the right from ‘ACTCGC’, so their edit distance is two. However,  these two strings differ on every position, so it is hard for neural network using phase 1 and 2 to find a good embedding such that after embedding these two strings are close to each other. Phase 3 solves this problem by padding. However, in most cases, the two strings are already aligned, then Phase 3 incurs cost because it may add more paddings and make the embedding worse.
> Empirically, we observe that phase 1+2 outperforms phase 1+2+3 for most datasets, except for hard cases. But adding phase 3 does not incur much cost (minor cost), so we recommend always adding phase 3, unless the users are extremely performance driven.
>
> Q: Why is a 2 layer GRU necessary and/or sufficient?
> A: Based on our experiments, 1-layer GRU gives inferior performance while 3-layer GRU gives similar results as 2-layer GRU. The network design intuition is mainly from the field of natural language processing. The sequence of characters in our problem resembles a sentence in NLP task, while each character in the sequence resembles a word in NLP. It is observed that if two sentences have similar meaning then GRU will learn similar embeddings for them. Therefore, we believe GRU (or other NLP network structures) is suitable for our task.
>
> Thank you for reading our responses. If you have further questions, we are happy to discuss.

---

### Official Review · AnonReviewer2 · 2019-10-22
**Official Blind Review #2**

**Rating:** 6

**Review:**

In this paper, the authors propose an approach for learning embeddings for strings using a three-phase approach. Each phase uses a different neural network to learn embeddings for strings such that the distance between strings in embedding space approximates the edit distance between the strings. A modest set of empirical results suggests that the proposed approach outperforms hand-crafted embeddings.

I found the presentation in this paper very disjointed and difficult to follow. While I believe (my interpretation of) the basic idea of this paper is interesting, I believe the current presentation significantly hinders readers from following the authors’ intentions.

Comments

The description of how the network structure and weights are “initialized” across the different phases is not clear. Different notation (f_1, f_2, f_3) is used for each network, but in reality, this is just the same network. However, the writing makes this very difficult to notice.

The authors introduce the CGK and CGK’ embedding algorithms, and then proceed to prove various properties about them. However, it is not clear to me how these theoretical properties are used by the neural network. From what I can tell, CGK’ is an alternative to CGK which reduces the output size relative to CGK (from 3n to at most 2n) while still ensuring exact reconstruction of the input. (I did not verify the proof in detail.) The authors then claim that this is helpful in the current context because it ensures the network parameters can be easily optimized. It is not clear to me what this means. (I guess that somehow using “CGK’ distance” makes training the model easier than using “CGK distance”.) Additionally, the experiments do not verify this claim empirically. So it is unclear whether using “CGK’ distance” helps in the context of learning embeddings.

It is really unclear to me whether the neural network outputs a continuous or a binary vector. In particular, Equations 5 - 8 all suggest that Hamming loss is defined on the outputs of the various neural networks (f_1, f_2, and f_3). The paper also refers to bits in the output of f_3. Later on, though, the paper mentions that the neural embedded strings are continuous vectors. While this could just be typos or inconsistent notation, considering that other parts of the paper do rely on binary representations, this makes the presentation very confusing.

It is unclear to me whether the can be (approximately) reconstructed from the embeddings. It seems that Theorem 1 suggests that the binary outputs of CGK’ can be decoded, but I cannot tell whether that extends to the embeddings.

It is unclear to me how positives and negatives are sampled for training in Phase 2, and also whether that impacts training.

The experimental results should include some measure of variance based on different train and/or test splits.

It seems as though the three phases could be rolled into a single multi-task learning problem in which the network is trained during a single phase.

Typos, etc.

The references are not consistently formatted.


**Experience Assessment:**

I have read many papers in this area.

**Review Assessment: Checking Correctness Of Derivations And Theory:**

I assessed the sensibility of the derivations and theory.

**Review Assessment: Checking Correctness Of Experiments:**

I carefully checked the experiments.

**Review Assessment: Thoroughness In Paper Reading:**

I read the paper at least twice and used my best judgement in assessing the paper.

---

> ### Author Response · Authors · 2019-11-10
> **Response to Official Review #2**
>
> Thank you for your detailed comments and suggestions! We have updated our paper accordingly.
>
> Q: Three phases are optimizing the same network, which is not explicitly mentioned
> A: Sorry for the confusion. We have updated the description in the introduction, and explicitly mention that we are optimizing the same network. Moreover, each phase is initialized using the final weights from the previous phase, and the first phase is using the default random initialization from PyTorch.
>
> Q: How theoretical properties of CGK/CGK’ are used by the neural network? And why is CGK’ better than CGK?
> A: CGK’ is for hard examples. Consider ‘ACTCGC’ and ‘CACTCG’. ‘CACTCG’ can be edited by shifting one character to the right from ‘ACTCGC’, so their edit distance is two. However,  these two strings differ at every position, so it’s hard for neural network that uses just phases 1 and 2 to find a good embedding such that after embedding these two strings are close to each other. However, CGK/CGK’ easily tackles this situation since it may randomly insert some characters into the string (see Algorithm 1), so that the two strings will be matched afterward. This is the main intuition behind our proof. We hope to use this alignment property so we apply CGK’ into our embedding, and optimize its parameters. Notice that CGK’ is better than CGK in the sense that CGK may repeat each character multiple times, but CGK’ only does it at most twice. Therefore, we only need to compare the two cases for each character, and use that information for optimization (see Algorithm 4).
>
> Q: In your experiments, did you verify that using CGK’ indeed helps?
> A: Phase 3 is the optimized version of CGK’ and empirically, experiments on synthetic hard case data set illustrate the improvement of utilizing phase 3. See Appendix A.3 for details.
>
> Q: The network outputs continuous or binary vector?
> A: Sorry for the confusion. The network outputs a continuous vector. We have fixed the confusing parts in the paper. And indeed, the analysis of CGK and CGK’ are based on binary vectors, but our optimized CGK’ (see Algorithm 3) can output continuous vectors.
>
> Q: Reconstruction of the embedding?
> A: Binary outputs of CGK’ can be decoded, but outputs of embeddings as a whole cannot be reconstructed.
>
> Q: How did you sample positive and negative data points?
> A: We randomly sample the positive pairs from top-5 edit distance and sample negative pairs as the top-30 distance in the embedded space. This approach is fairly standard for triplet loss, see e.g. [1, 2].
> [1] Distance metric Learning for large margin nearest neighbor classification.
> [2] Spreading vector for similarity search
>
> Q: Can you include some measure of variance based on different train and/or test splits
> A: Sure. We have tried different test queries and due to time constraint, we have only updated the results of Neural Phase 1+2+3 in Table 5. We will update other results as well in the next version of our paper.
>
> Q: Why use 3 phases, not just a single phase?
> A: We have tried to train phase 1 and phase 2 together as a multi-task problem, but the empirical results we obtained were not as good. This is because phase 1 first scales and approximates the edit distance, and then phase 2 further captures the relative similarity, so the order of phase 1 and phase 2 seems to matter. Furthermore, phase 3 cannot be jointly trained with phase 1 and phase 2, because in phase 3 we are also optimizing the CGK’ algorithm, which uses approximation to the gradients (see Algorithm 4). This phase is more like a fine tuning step.
> Moreover, even if all phases can be trained together, one needs to tune the final loss function that combines all the losses together. Our current three phases training avoids this tuning problem because we just train the network until convergence for each phase, without using any hyperparameters.
>
> Q: Reference format not consistent
> A: Thanks for pointing them out! We have adjusted the format.
>
> Thank you for reading our responses. If you have further questions, we are happy to discuss.

---

> > ### Comment · AnonReviewer2 · 2019-11-15
> > **RE: Author response**
> >
> > I have read the other reviews and authors' responses; I also briefly looked into the updated paper. These have clarified a number of important details, and I have updated my score to "weak accept" as a result.
> >
> > To me, there is still a bit of a disconnect between developing the CGK' algorithm, proving various decoding properties about the binary vectors it creates, and then learning continuous vectors where the proofs no longer hold. Would it be possible to show some sort of approximate "rounding" result (e.g., if we round the continuous vectors to binary ones, we recover the original sequence with some bounded likelihood)?

---

> > > ### Author Response · Authors · 2019-11-15
> > > **your comment**
> > >
> > > Thank you for your response! The question you ask (about "rounding") is very interesting but it appears to be quite non-trivial. We will investigate it over the next few weeks.

---

> ### Author Response · Authors · 2019-11-13
> **Further Update of Variance Based on Different Train and/or Test Splits**
>
> We have further added the standard deviation of Table 2 and Table 5 by testing with five different query sets, each of which contains 100 randomly selected strings. Thanks!

---

### Official Review · AnonReviewer3 · 2019-10-23
**Official Blind Review #3**

**Rating:** 3

**Review:**

Authors propose a three phase learning schedule to find embedding vectors for sequences. The goal is to have the euclidean distance of embedding vectors mimic the edit distance of input sequences. Given such embedding one can perform faster approximate nearest neighbor search in compare to calculating pairwise edit distances.
They use a RNN to output a real value per sequence step. At first they pretrain with the absolute difference of euclidean distance and edit distance. Then they fine tune with a triplet loss, such that the difference in euclidean distances be larger than the difference in edit distances. Finally they modify the embeddings with a stochastic, differentiable algorithm such that they can get guarantees for generalization.

Unfortunately, the manuscript is not well written. There is a high chance of misunderstandings. What I gather from the experiment section is that their model is trained on the whole corpus. During training repeatedly trains with absolute loss on pairwise edit distances. During inference random 100 of those same sequences that has been trained on are selected to compare with the rest. If this is true, I fail to grasp the point of this paper. Since during training you have effectively calculated all the pairwise edit distances. There is no generalization happening. This paper has effectively memorized the edit distances of some sequences.

It seems that only phase 3 (cgk') is designed to have any accuracy on unseen sequences, and experiments show that it underperforms the original cgk.

If this is not true and indeed they are training for example on one half of the corpus and the 100 query + base are unseen during training I am willing to increase my score. Given the added clarification in the paper.

Again assuming that this is not just memorization:

Why eq 5 (regression loss) is the absolute value? It means that you will never get closer than lr/2 to the optimal point, where as with a least squares loss your gradients get smaller when you get closer.

How are the negative sample, positive samples selected for an anchor? Are they just two random points, is there any importance sampling happening?

Why during phase 2, the phase 1 loss is stopped? There is no intuition, justification in the paper. Why the loss is not eq 5 + eq 6 during the whole training?

How are you optimizing? SGD I assume? How are you selecting hyper-parameters, such as learning rate? Is there any validation set?

Is there a typo in eq 7? The text says "we calculate the absolute loss as in Phase 1 to optimize our embedding network f3" but eq 7 is on f'(3). Are you backpropagating through algorithm 3 toward embeddings or just toward thres as in algorithm 4?

Currently, given the poor quality of the write up, the merit of the idea and the experiments is not clear.

Related work: LSDE: Levenshtein Space Deep Embedding for Query-by-string Word Spotting

**Experience Assessment:**

I have published one or two papers in this area.

**Review Assessment: Checking Correctness Of Derivations And Theory:**

I assessed the sensibility of the derivations and theory.

**Review Assessment: Checking Correctness Of Experiments:**

I carefully checked the experiments.

**Review Assessment: Thoroughness In Paper Reading:**

I read the paper thoroughly.

---

> ### Author Response · Authors · 2019-11-10
> **Response to Official Review #3**
>
> Thank you for your detailed comments and suggestions! We have updated our paper accordingly.
>
> Q: Are you memorizing the dataset?
> A: No, we are not memorizing the dataset, sorry for the confusion! In our experiment, we use less than 2% data points in the whole dataset for training. We have updated the paper in Section 5.1 to make it clear.
>
> Q: Why use absolute loss instead of least square loss?
> A: In our implementation, we have adopted learning rate decay so the learning rate becomes smaller with epochs increasing, which solves the lr/2 distance problem. Empirically, we observe that absolute loss has similar performance compared with least square loss. But you are right, if one uses fixed learning rate, it makes more sense to use least square loss.
>
> Q: How did you sample positive and negative data points?
> A: We randomly sample the positive pairs from top-5 edit distance and sample negative pairs as the top-30 distance in the embedded space. This approach is fairly standard for triplet loss, see e.g. [1, 2].
> [1] Distance metric Learning for large margin nearest neighbor classification.
> [2] Spreading vector for similarity search
>
> Q: Why during phase 2, the phase loss is stopped?
> A: We have tried to train phase 1 and phase 2 together as a multi-task problem, but the empirical results were not as good. This is because phase 1 first scales and approximates the edit distance then phase 2 further captures the relative similarity, so the order of phase 1 and phase 2 matters. We have also tried to first train phase 1 then eq5 + eq6 together, but the results were also not as good, potentially because it’s not easy to tune the final loss function that combines different losses together.
>
> Q: How are you optimizing?
> A: We use Adam optimizer and 4:1 train: validation split for cross-validation and tune the hyper-parameters.
>
> Q: Is there a typo in eq7?
> A: Thank you for pointing it out. This is not a typo, but the description is indeed confusing. We have updated it in the new version, and hopefully it’s now better.
>
> Q: related work: LSDE: Levenshtein Space Deep Embedding for Query-by-string Word Spotting
> A: Thanks for pointing out this related work. We have cited it in our paper.
>
> Thank you for reading our responses. If you have further questions, we are happy to discuss.

---

### Public Comment · ~Xinyan_Dai1 · 2019-10-25
**Could you please provide source code?**

Hi, it's an interesting idea. Is it possible to share the source code using an anonymous link(e.g. putting the source code on gofile)? Thanks.

---

> ### Author Response · Authors · 2019-10-25
> **Anonymous Code Available**
>
> Thanks for your interest in our work! Code is available here https://drive.google.com/drive/folders/1FuyzFq7vIh9biTY3YFRgFRgfRCehng5C?usp=sharing

---

### Decision · Program_Chairs · 2019-12-19

**Decision:**

Reject

**Comment:**

This paper presents an approach to improving the calculation of embeddings for nearest-neighbor search with respect to edit distance.

Reading the reviews, it seems that the paper is greatly improved over its previous version, but still has significant clarity issues. Given that these issues remain even after one major revision, I would suggest that the paper not be accepted for this ICLR, but that the authors carefully revise the paper for clarity and submit to a following submission opportunity. It may help to share the paper with others who are not familiar with the research until they can read it once and understand the method well.

I have quoted Reviewer 3 below in the author discussion, where there are some additional clarity issues that may help being resolved:

----------

Some specifics are clear now with their new edition.
* The [relationship between] cgk' & cgk not as clear as it could be. For example the algorithms are designed for bits. So one should assume that they are applying it on the bits of the characters. But this should be clarified in the manuscript.
* Also still backpropagating through f' is not clear to me.
* And in the text for inference they still say: "We randomly select 100 queries and use the remainder of the dataset as the base set" which should be "the remainder excluding the training set" or "including?".